# Reproducibility and Data Storage for Active Learning-Aided Systematic Reviews

Peter Lombaers [1,2], Jonathan de Bruin [3] and Rens van de Schoot [1,*]

1  Department of Methodology and Statistics, Faculty of Social and Behavioral Sciences, Utrecht University, 3584 CS Utrecht, The Netherlands; p.j.lombaers@uu.nl or peter@idfuse.nl
2  IDfuse, 3526 KS Utrecht, The Netherlands
3  Department of Research and Data Management Services, Information Technology Services, Utrecht University, 3584 CS Utrecht, The Netherlands; j.debruin1@uu.nl
*  Correspondence: a.g.j.vandeschoot@uu.nl; Tel.: +31-302534468

**Featured Application: Increasing reproducibility for active learning-aided systematic screening is essential and our checklist can be used to evaluate reproducibility and data efficiency of software.**

**Abstract:** In the screening phase of a systematic review, screening prioritization via active learning effectively reduces the workload. However, the PRISMA guidelines are not sufficient for reporting the screening phase in a reproducible manner. Text screening with active learning is an iterative process, but the labeling decisions and the training of the active learning model can happen independently of each other in time. Therefore, it is not trivial to store the data from both events so that one can still know which iteration of the model was used for each labeling decision. Moreover, many iterations of the active learning model will be trained throughout the screening process, producing an enormous amount of data (think of many gigabytes or even terabytes of data), and machine learning models are continually becoming larger. This article clarifies the steps in an active learning-aided screening process and what data is produced at every step. We consider what reproducibility means in this context and we show that there is tension between the desire to be reproducible and the amount of data that is stored. Finally, we present the RDAL Checklist (Reproducibility and Data storage for Active Learning-Aided Systematic Reviews Checklist), which helps users and creators of active learning software make their screening process reproducible.

**Keywords:** systematic review; meta-analysis; active learning; transparency; open science; data storage; reproducibility



## 1. Introduction

The number of scientific papers on any topic is skyrocketing, and the world's scientific output doubles every nine years [1]. For this reason, systematically and rapidly screening many texts for relevance is becoming increasingly important. The number of systematic literature reviews is growing rapidly [2]. Systematic searches form the basis for synthesizing the state-of-the-art in a particular field and might be used not only for systematic, scoping, narrative, mapping, or even umbrella systematic reviews, but also for meta-analyses, diagnostic test accuracy, or network meta-analysis [3]. The process of systematic searching entails several explicit and reproducible steps, as outlined in the PRISMA (Preferred Reporting Items for Systematic Reviews and Meta-Analyses) guidelines [4]. Such efforts help to make the result of a systematic search FAIR: findable, accessible, interoperable, and reusable [5].

Developing a dataset of potentially relevant papers from a systematic search is an iterative process aimed at balancing recall and precision [6], including as many potentially relevant studies as possible (recall) while simultaneously limiting the total number of studies to be screened (precision). Since a single literature search can easily result in

thousands of publications that must be read and screened for relevance, literature screening is extremely time-consuming [7]. Artificial intelligence can help to speed up the process of searching through large amounts of text data. Machine-supported pipelines have been developed that assist in finding relevant texts for search tasks; for overviews, see [8–12]. A well-established approach to increase the efficiency of title and abstract screening is screening prioritization [13,14] via active learning (AL) [15]: a constant interaction between a human and a machine; for an explanation, see Box 1 and Figure 1. Active learning is extremely effective in screening large amounts of textual data [9,14,16–19] and is success-fully implemented in screening software like Abstrackr [17], ASReview [9], Colandr [18], FASTREAD [16], Rayyan [20], RobotAnalyst [21], Research Screener [22], DistillerSR [23], and robotreviewer [24]. Note that this is different from the classical notion of active learning in machine learning; see Box 2 for an explanation of the differences.

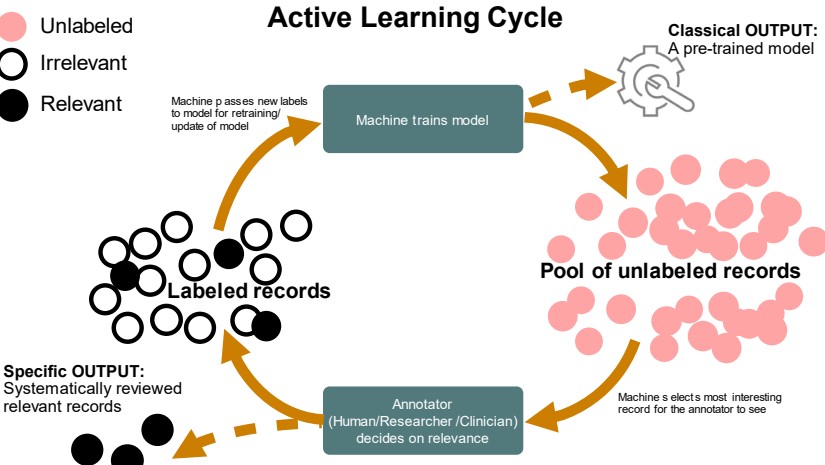

**Figure 1.** The human-in-the-loop (HITL) active learning cycle with two types of output: on the top right, the trained model, which is the main output in the classical active learning; and at the bottom left, all relevant papers, which is the main output of researcher-in-the-loop (RITL) active learning.

**Box 1.** Active learning explained.

Researcher-in-the-loop active learning (RITL) is a form of active learning (AL) where the follow-ing three criteria need to be satisfied [9,25]:

- The primary output is a list of labeled records, and the machine learning model is only secondary.
- The annotator needs to have seen the relevant records at the end of the active learning process.
- The active learning process should be transparent and reproducible.

The process starts with a database search to obtain a dataset of unlabeled records containing meta-data such as titles, abstracts, and keywords of scientific papers. This is followed by constructing an initial labeled *training set*, where the labels are provided by the *annotator*. This consists of at least one *labeled relevant* and *labeled irrelevant* record. Next, an active learning model needs to be selected, including a *feature extraction* technique, a *classification algorithm* (i.e., a machine learning model), a query strategy, and a balancing strategy to deal with the extreme imbalance between the number of relevant and irrelevant records in the data. Then, the active learning cycle starts (see also Figure 1):

1. The chosen model is trained on the labeled records, and the model is used to produce *relevance scores* for all unlabeled records.
2. The annotator samples a record from the unlabeled records, for example, the record with the highest relevance score (certainty-based sampling [26]).
3. The annotator screens this record and provides a label, i.e., relevant or irrelevant, and goes back to step 2. If there are enough new labeled records, step 1 is also triggered.

This cycle is repeated until the annotator thinks they have seen all relevant records. The list of relevant records is the output of the system and will be used in the subsequent data extraction phase. The goal is to save time by screening fewer records than exist in the entire pool because the active learning model puts all of the most likely relevant records at the front of the list.

**Box 2.** Differences between classical active learning and RITL.

> Classical active learning refers to a set of techniques where a model is trained on a subset of the available data, and then actively selects additional data points for which it requests labels. The goal is to improve the performance of the model while minimizing the amount of data that needs to be annotated. The main output is a trained model that can be used on unseen records, and the annotated data is a byproduct. It can be particularly useful in cases where obtaining new labels is expensive, for example, because a (human) expert needs to look at the unlabeled records. Similarly, in the RITL set-up, obtaining new labels is an expensive process. However, unlike in the classical set-up, the main output is the set of labeled records, not the trained model.
>
> An example of classical active learning would be training a model to classify images. The model selects the images for which a human annotator should provide labels. The model selects the images from which it can learn the most, so that the annotator needs to label as few of them as possible. In the end, the trained model can be used to classify new unseen images. The reproducibility of the process enhances our trust in the process; it may be more important, however, that others can reuse the results and data to improve or analyze the model.
>
> A typical application of RITL would be to find all relevant articles for determining a dosage guideline for a medical drug. First, databases are searched for possible articles, and these articles are then screened by a human annotator. Instead of annotating all articles, a model is iteratively trained to suggest the most relevant articles the annotator should look at first. The list of relevant articles is the main output of the system. It is important that the human annotator sees all relevant articles because we do not want the model to make the final decision about whether an article is relevant. The main reason we want the process to be transparent and reproducible is to increase the trust in the dosage guideline.
>
> Obviously these differences between classical and RITL active learning have implications about what it means to be reproducible, and what data is important to store. Clearly in the classical setting it is more important to store information regarding the training of the model, while in the RITL setting, the labeling decisions have higher priority. As we explain in Section 3, the model data are typically larger than the labeling data. In the classical setting, the larger model data are the primary output. In contrast, in the RITL setting, the labeling data are the primary output and storing the secondary model data means a large increase in storage size. Therefore, data storage size is more of a consideration for RITL.

While the active learning pipeline can significantly reduce the time spent on systematically screening, it also presents challenges regarding transparency and repro-ducibility. At the end of the AL-aided screening process, there is a set of seen and la-beled papers and a set of unseen papers without labels. To understand why the anno-tator did not see these precise papers, we need to have insight into the model's deci-sions at any time in the process. This presents theoretical and practical challenges. Although the field of explainable Artificial Intelligence is gaining lots of attention [27], it can be difficult or even impossible to explain how a system comes to its exact deci-sion, especially when neural nets are used [28].

But before explainable Artificial Intelligence can even be applied to an active learning-aided screening pipeline, the model and the data the model produces should be stored. This presents difficulties for two reasons. Firstly, text screening with active learning is an iterative process, but the labeling decisions and the training of the active learning model can happen independently of each other in time. So it is not trivial to store the data from both events in such a way that you can still know which iteration of the model was used for each labeling decision. It is not necessarily the case that a new model will be trained after each labeling decision. Secondly, many iterations of the active learning model will be trained throughout the review, producing an enormous amount of data (think of many gigabytes or even terabytes of data), and machine learning models are continually becoming larger using even more model parameters [29]. Together this can add up to an undesirable amount of data when naively storing all the data produced at every iteration of the active learning pipeline. Also, many studies fail to report enough detailed information about the datasets and machine learning algorithms used, limiting the reproducibility of the studies assessed [30,31].

Thus, in AL-aided systematic screening, there is a tension between the desire for reproducibility and transparency on the one hand, and the practical difficulties of storing and interpreting machine learning models on the other hand. More data must be saved if a more complete and straightforward reproducible result is wanted. In classical systematic reviews, there are clear guidelines on what to report regarding the output and the process of creating the review [4]. However, there is no such consensus for AL-aided systematic reviews, and PRISMA only offers general recommendations [32]:

*Specify the methods used to decide whether a study met the inclusion criteria of the review, including how many reviewers screened each record and each report retrieved, whether they worked independently, and if applicable, details of automation tools used in the process.*

In the expanded checklist (source: https://prisma-statement.org//documents/PRISMA_2020_expanded_checklist.pdf (accessed on 25 April 2024)), this is elaborated further:

*Recommendations for reporting in systematic reviews using automation tools in the selection process:*

- *Report how automation tools were integrated within the overall study selection process. [...]*
- *If machine learning algorithms were used to prioritize screening (whereby un-screened records are continually re-ordered based on screening decisions), state the software used and provide details of any screening rules applied.*

However, as we demonstrate in the current paper, the updated PRISMA guidelines are not enough when using AL-aided pipelines. In AL-aided screening, this is more difficult because the annotator sees only part of the data. Suppose two people start reading the same dataset. In that case, the active learning model might suggest different records to read because they started with a different training set or selected a different model, there was difference of opinion between the annotators, or because of one of many more possible reasons. As a result, they might end up with different results (i.e., a different set of seen and labeled records), even though the dataset and inclusion/exclusion criteria were identical. In the worst-case scenario, relevant records remain in the unseen set, for example, because the model did not recognize them or because one of the records was mislabeled by the annotator. In order to understand what happened in such a situation, resolve any conflicts between the two decision sets, and have confidence in the process of AL-aided screening, the process must be reproducible.

The current paper starts with a general discussion of reproducibility in the context of AI-aided systematic screening. After that, we fully focus on reproducibility for the phase of systematically screening records. We do not look further at other phases of a systematic review, such as database searching or data extraction. AI tools can be used in these phases, and reproducibility is also important there, but it is a topic for another paper. We also do not look at topics such as the performance of AI-aided screening or biases in AI-aided screening. These, in fact, are motivations for making screening data reproducible and accessible.

We give a detailed description of all the steps taken during screening. Then we look at the data generated by the human screener and by the model during each step, and we indicate how large the data storage size is relative to the size of the input dataset. For each piece of data, we try to assess its importance for the overall reproducibility. We provide a generic data storage framework that uses these insights to minimize storage size while keeping it as reproducible as possible, specifically aimed at RITL. We describe the implementation of the framework in the open-source software ASReview [33]. Finally, we propose a new checklist: the Reproducibility and Data storage Checklist for Active Learning-Aided Systematic Reviews, the RDAL checklist for short. It is meant to help users and creators of active learning screening software to answer the question: What do I

minimally need to store so that others can have confidence in the results, and are able to reuse them?

## 2. Reproducibility in the Context of Systematic Screening

Reproducible research is not an easy concept to define. In very broad terms, the meaning is to describe the data, methods, and results of the research in such a way that others can start with the same data and use the same methods to arrive at the same results. What 'same', 'data', 'methods', and 'results' mean depends very much on the context of the research and the goal of reproducibility. The confusion around the word 'reproducibility' is confounded by the fact that there is also the word 'replicability', and the meaning of these two words can be completely opposite depending on the context. See Peng and Hicks [34] for a review and Barba [35] for an overview of the usage of the term reproducibility.

A systematic review is a process with many steps, and in all of those steps one can debate about what is necessary to store for it to be reproducible. In every step, the answer will depend on the goal of reproducibility. In the case of a systematic screening, the main goal is to find all texts relevant to a research question. Therefore, the main goal of reproduction should be to understand and verify each step that led to this list of texts. This builds trust in the process as a whole. In the case of AI-aided screening, there is the extra goal of understanding and verifying the AI-method used and the specific implementation used. Then there are secondary goals: allowing easy reuse of the data for further analyses of the results, producing a trained AI model, and improving the AI method. In the case of classical active learning, we would have a different primary goal, namely, to produce a model that can accurately label texts. Of course, this means that the model training steps of the process become more important for reproducibility. Thus, we see that the goal influences what data we want to store.

Another important consideration is the ease of reproducibility. Some things are very easy to store and reproduce, such as a list of labels 'included' or 'excluded' for each record in the screening phase. Other information is more difficult to store, such as the state of an AI model after training it on certain records. We could store all the information on the model, but this might become very large, especially with large, modern machine learning models. On the other hand, we can store only the information needed to train the model again, but then it will take more time to reproduce the model; or, it will even be impossible to reproduce the model for someone that does not have access to sufficiently powerful hardware to run the model. In such a case, it would be reproducible for certain people, but irreproducible for others.

We take a detailed look at all the steps in the screening phase of a systematic review. For each of these steps, we look at the data that goes in and comes out. With this information at hand, and with our main goal of reproducibility in mind, we give recommendations on what data should be stored, and how it should be stored, in order to strike a good balance between ease of reproduction and data storage.

Note that we focus on those ingredients relevant to the active learning aspect of the pipeline, thereby ignoring other important issues like search strategies, deduplication, abstract versus full-text screening, and inter-rater reliability. Such topics are already clearly described in textbooks (e.g., [32]) and the PRISMA guidelines. We also do not consider using artificial intelligence tools in other steps than in the screening phase.

### 2.1. Defining Three Phases

We can identify three main phases in a systematic review; see Figure 2:

1. the pre-processing phase;
2. the screening phase;
3. the post-processing phase.

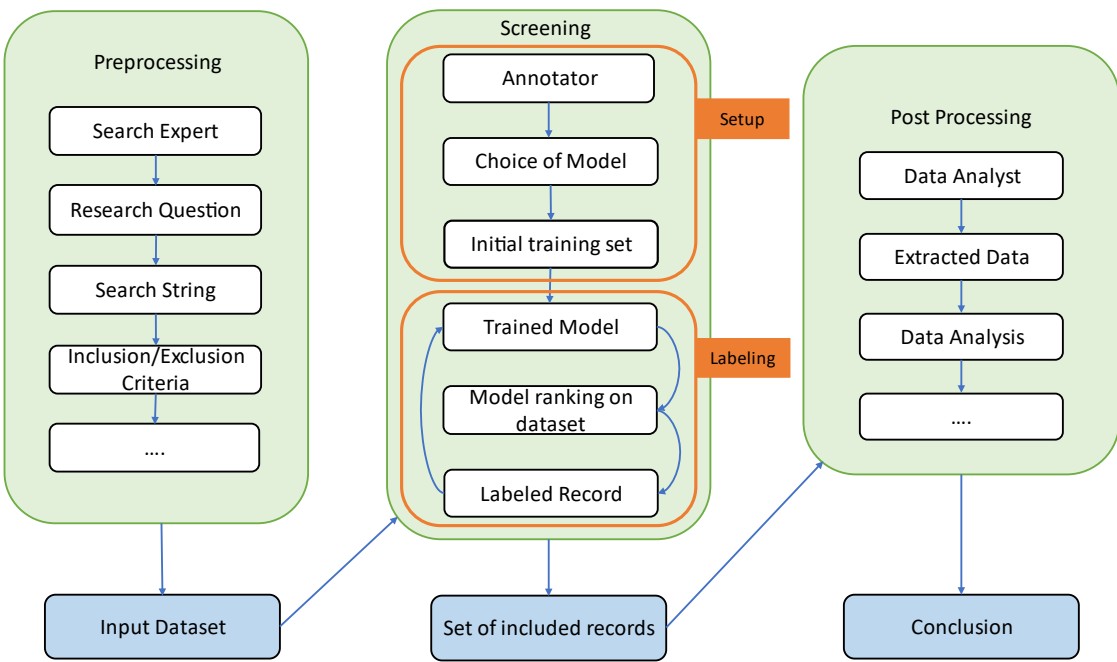

**Figure 2.** Main steps in a systematic review using active learning.

The pre-processing phase results in an input dataset containing the information for the screening phase. To create such a dataset, the researcher formulates a research question and determines the inclusion and exclusion criteria for selecting relevant records. Then, they create a dataset that contains all records that might be relevant to the question. Typically this is conducted using database searches in multiple databases, after which the results are combined and deduplicated. The input dataset is a set of records containing meta-data of, for example, scientific papers (i.e., titles, abstracts, and persistent identifiers like the DOI).

In the screening phase, it is determined which of the records in the input dataset are relevant based on a predefined set of inclusion and exclusion criteria. The screening phase is an iterative and synchronous process, except for the training of the active learning model, which can happen asynchronously; see Figure 3. In each iteration, a model is trained on the labeled records, starting with the initially selected prior knowledge. The trained model then predicts relevance scores for the unseen records. It ranks these records from high to low. At the same time, the annotator reads the records in the current ranking order and assigns each record a label. The new label(s) are then used to train a new model. After each labeling decision, the size of the training data increases.

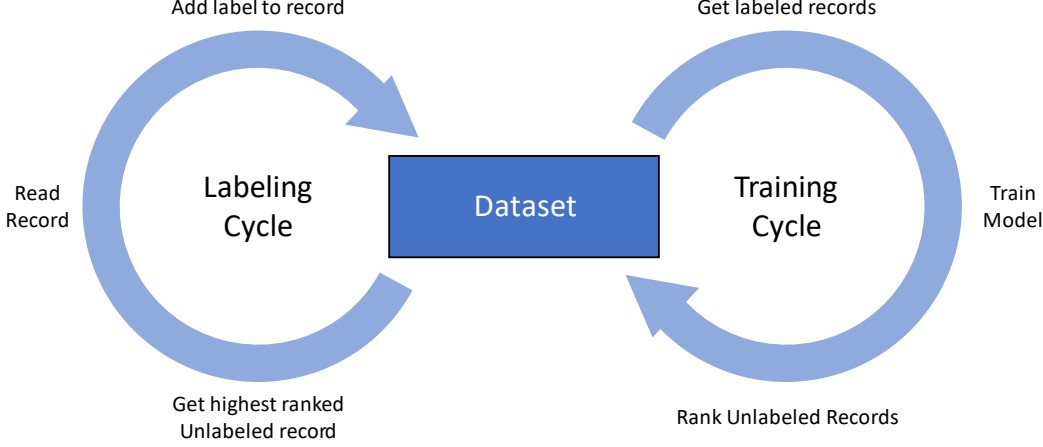

**Figure 3.** The two cycles of labeling and training.

Note that both these processes (training a model and labeling records) are continuous and can be parallel to each other; at any moment, a new model can be trained using the currently labeled records, and at any moment, the annotator can take a new record from the pool of ranked, unlabeled records and assign it a label. The annotator and the model do not need to wait for each other. Of course, it is an option to implement the labeling and model training in a serial fashion, one after the other, but this is a priori not necessary and is inefficient. The user always wants to be able to request a new record instead of having to wait for the model to finish with training. Which record is selected depends on the chosen query strategy. It can be the record with the highest relevance score, called certainty-based sampling [26], which is most often used for HITL active learning.

This asynchronicity is a departure from the classical active learning setting. In that setting, the model can request which records the annotator should label. Labeling extra records from an old iteration of the model could mean wasted time by the annotator since the new iteration of the model might request different records from which it can learn more. Therefore, making the model training and the labeling phases asynchronous can mean that resources are wasted. In the RITL setting, this is not a problem since the set of labeled records is the main output anyway, and thus an asynchronous set-up means there is less downtime for the annotator.

Once the screener decides to stop labeling, the relevant records can be exported for post-processing (e.g., data extraction). We define the output of this phase as the set of relevant records. The included records from the output dataset are used in the post-processing phase. In the post-processing phase, the data of the selected relevant records are extracted and analyzed in order to answer the research question. Thus, the output of this phase is the conclusion of the study.

### 2.2. Reproducibility in the Three Phases

In Goodman, Fanelli [36], the authors try to clarify the reproducibility terminology by distinguishing between three different kinds of reproducibility depending on the goal: 'method reproducibility', 'results reproducibility', and 'inferential reproducibility'. We can relate this to the different steps of the screening phase outlined in Figure 2. The strictest form of reproduction would be that given an identical input dataset, annotator, choice of model, and initial training set, from which we obtain identical trained models, model rankings, and labeled records, seen in identical order. This form of reproduction checks that repeating the same steps leads to the same results at every point in between. It tells us that the system is behaving predictably. This is what is meant by 'method reproducibility'.

We could also keep everything fixed but allow for a different annotator. In that case, we can check that another person can obtain the same results using this system. If the annotator makes the same decision for each record, the end result should be the same. However, this will not always happen since annotators might disagree about whether certain records should be included or they make mistakes. This form of reproducibility is the simplest form of 'results reproducibility' from Goodman, Fanelli [36].

A more functionally oriented definition of reproduction would be to demand that the input dataset is identical but allow for variation in the other steps as long as the final set of included records is the same. This procedure checks that the final results are not dependent on the choices made for the active learning system. Of course, this is specific for RITL active learning (see Box 1), where the set of labeled records is the primary output. In the case of classical active learning, we would need to demand that the trained models are identical or at least very similar.

Lastly, we could take an even broader view and argue that the set of included records is also allowed to vary as long as the conclusion coming out of the post-processing stage is still the same. This perspective encourages a broader view of reproducibility, where the focus is not only on individual records but also on the robustness of the overall conclusions when subjected to different analytical conditions. This is called 'inferential reproducibility' [36], suggesting that a robust conclusion should ideally be stable across different subsets of data,

provided the subsets are representative and substantial. This does not imply that missing records are inconsequential, but rather that the conclusions of a well-conducted review should be resilient to the variations that might arise from different systematic screening approaches. Reproducing a systematic screening phase in this way shows whether missing a specific record is essential for the final conclusions; for an example, see [37].

### 3. Data in AL-Aided Screening

In this section, we describe what data is generated during each step in the screening phase; see also Table 1 and Figure 2. In the screening phase, we can differentiate between the set-up and the labeling phase.

**Table 1.** Data types and sizes in active learning aided systematic screening.

| Type of Data | Stored Every Iteration? | Size Depends on Dataset? | Total Size S/M/L (Small/Medium/Large) | Essential to Store? |
|---|---|---|---|---|
| Input data | | | | |
| Meta-data like titles and abstracts | N | Y | S | N |
| Persistent object identifiers, like DOI | N | Y | S | Y |
| Set-up phase | | | | |
| Meta-data of the project (annotator, title, description, etc.) | N | N | S | Y |
| Software plus version | N | N | S | Y |
| Model Settings (feature extraction, classifier, query strategy, balance strategy) | N | N | S | Y |
| Feature Matrix | N | Y | M/L (M if the number of features is small) | Y |
| Records used for training the model plus their labels | N | N | S | Y |
| Random seed values | N | N | S | Y |
| Intended stopping rule for labeling | N | N | S | Y |
| Intended stopping rule for model training | N | N | S | N |
| Screening Phase—data produced by the annotator | | | | |
| Order of labeling | Y | N | M | Y |
| Labels | Y | N | M | Y |
| Time of labeling | Y | N | M | Y |
| Notes | Y | N | M | Y |
| Changed Decisions | N | N | S | Y |
| Screening Phase—data produced by the model | | | | |
| Model settings per iteration | Y | N | M | Y (in case of model switching or training multiple models) |

**Table 1.** *Cont.*

| Type of Data | Stored Every Iteration? | Size Depends on Dataset? | Total Size S/M/L (Small/Medium/Large) | Essential to Store? |
|---|---|---|---|---|
| Model parameters per iteration | Y | Y | M/L (depends on the size of the feature space) | N |
| Training set | Y | Y | M/L (M if only size of the training set) | Y |
| Relevance scores | Y | Y | L | N |
| Ranking | Y | Y | L | N (only the ranking of the last iteration) |
| Model training time | Y | N | M | N |

### 3.1. Data in the Set-Up Phase

The set-up phase prepares the data that is necessary to train the first iteration of the machine learning model on the input data (i.e., the result of the pre-processing phase). The user chooses the software (with a specific version). Also, a stopping rule for when to stop labeling records should be decided upon. This can be far from trivial, though, as discussed elsewhere in great detail, e.g., [38].

Moreover, the four components of the active learning model need to be selected: (1) The type of machine learning model with its hyperparameters. Random seeds should be used and stored to make any probabilistic process in the model reproducible. (2) The dataset containing text is transformed into a format the model can understand using a feature extraction method, and this feature matrix should be stored. Suppose a user switches to a different active learning model during the screening phase consisting of a different feature extraction technique. In that case, multiple matrices need to be stored. (3) Also, a balancing strategy needs to be selected to deal with the sparseness of the relevant records. For an example of a balancing strategy, see Appendix 3 of Ferdinands, Schram [39]. (4) Finally, a query strategy needs to be selected, determining which record will be selected by the model to be shown to the annotator.

Similarly to the stopping rule for labeling, a stopping rule for model training can be selected. While it might seem logical to continue training new models until the end of the screening process, this probably is not necessary. After enough labels have been provided, a new label will not dramatically change the model. Therefore, to continue training new models might be a waste of energy. However, this model training stopping rule is less important than the labeling stopping rule, because the labeled data are the primary output.

Furthermore, a training set for the model needs to be created by labeling a set of records as relevant and irrelevant based on prior knowledge (e.g., for the relevant set), or screened randomly (e.g., for the irrelevant set). The minimal required prior knowledge for the training data is at least one label '0' and one label '1', but any amount of prior knowledge for the training data can theoretically be used. It depends on the software implementation and what the options are.

The data produced in the set-up phase will only have to be stored once at the start of the review. Moreover, except for the feature matrix, the data size does not depend on the size of the input dataset. Therefore, the data generated during the set-up phase will not become too large relative to the input dataset.

### 3.2. Data in the Labeling Phase

The data produced during the labeling phase can be divided into data generated by the annotator and data generated by the model, where the data generated by the annotator includes the labeled records, with the corresponding label, the time of labeling, any notes made by the user, any changed decisions, and skipped records. This needs to be saved once for each labeled record; therefore, the data size will be in the same order

of magnitude as the dataset itself. The data generated by the model includes the type of model (users might switch to a different model during screening, or multiple models are trained simultaneously), any settings used specifically for the current iteration of the model, including training times, the parameters of the model, relevance scores assigned by the model to each record, and the ranking of the records in order of relevance. Assuming one model is trained each time, the number of trained models will be closely related to the number of labeled records, and thus, it will be linearly related to the size of the dataset. Although the data size of the model typically does not depend on the size of the dataset, it can still be very large. The size of modern neural networks for natural language processing can easily be several hundreds of megabytes or even gigabytes. However, for each model, the data size also depends on the size of the dataset; there will be one relevance score for each record in the dataset for each iteration in the model. Together, this means that the data size of all the generated relevance scores will depend quadratically on the size of the dataset.

## 4. A Trade-Off between Reproducibility and Data Storage

In practice, there is a trade-off between data size on the one hand and ease of reproducibility on the other hand. By using extra data storage, we can make it easier to reproduce the state of the screening at a certain iteration or moment in time. However, when these data (model parameters, relevance scores, etc.) become much larger than the primary output (a list of labeled records), this becomes a problem. Therefore, we propose only to store some essential components to make the output of the AL-aided pipeline transparent and reproducible; for an overview, see Table 1.

At one extreme of the spectrum between data size and time, the least amount of storage capacity involves only storing the starting point (input dataset, initial training data, model settings, seed values, and inclusion/exclusion criteria). Someone reproducing the results can import the data and select the same model, and by screening the records in the order they are suggested and applying the inclusion criteria, they should obtain the same set of labeled records and the same trained active learning models. This will cost a large amount of time, though; the annotator needs to read all the records, and the active learning system needs to train all the iterations of the model again.

Moreover, some accuracy will be lost. Humans make mistakes when reading and labeling records [7], and making one different labeling decision will result in a different set of records shown to the annotator. Also, many machine learning algorithms use (pseudo)random number generators to simulate probabilistic behavior. To make these algorithms deterministic, the random seed needs to be stored at the start. However, many algorithms designed for a graphics processing unit (GPU) are not deterministic due to the many parallelized computations that occur simultaneously. In recent years, support for deterministic training of machine learning models on GPUs has increased [40,41]. In the end, the results (i.e., the set of seen and included records) might be replicated, but the entire process is not reproduced precisely because of both human errors and probabilistic models.

At the other extreme of the reproducibility spectrum, one can try to store absolutely everything. This means storing every iteration of the model, with all its model parameters and the relevance scores that it produced for every record. In most cases, this becomes infeasible and, in fact, undesirable; if the data size becomes too big, it will be more difficult for other people to use the data and investigate its contents. For example, storing the relevance scores of a thousand trained models on a dataset of a thousand records takes approximately 8 MB, and the scores of ten thousand models on ten thousand records take approximately 800 MB. With every increase in the size of the dataset by a factor of 10, the data storage size increases by a factor of 100. When the dataset has in the order of a million records, the data size becomes in the order of terabytes, and this is just for storing the relevance scores; imagine how much storage capacity is needed to store the entire output of the model for each iteration.

## 5. Data Storage Recommendations for Systematic Screening

Now that we have a description of the different steps in the screening process and the data generated during each step, we can give recommendations on what to store in each step. We describe the essential components needed to be stored for an optimal trade-off between transparency, reproducibility, and data storage capacity; also see the last column in Table 1.

### 5.1. Input Data

Sharing the search query, as requested by PRISMA, has become the standard in the field. For example, out of 117 systematic reviews published at Utrecht University in 2020, 91 published the entire search query [42]. Haddaway, Rethlefsen [43] suggested a data structure for transparent and repeatable reporting of bibliographic searching. We advise users to follow such recommendations. A standardized way to report a bibliographic search might sound desirable, but reproducing such a search years later is problematic; journals are added to or omitted from databases, search functionalities change, and the meta-data are continuously updated. A solution would be to store the entire dataset resulting from the search; in the study, only 5 out of the 117 reviews did so.

However, not all meta-data can openly be shared because abstracts might fall under the strict copyright of the publisher, which limits the application of global text and data mining research [44]. Until publishers adhere to the plea for releasing such restrictions, we propose storing a list of persistent digital object identifiers, like the DOIs. Then, via automated tooling, using Lens [45] or OpenAlex [46], it is possible to compose the full meta-data via DOI-matching. Note that storing the DOI is also not perfect since the meta-data associated with a DOI can change, for example, via an updated mesh term or keyword, but the object it refers to is persistent.

### 5.2. Set-Up Phase

Everything in the set-up phase should be stored entirely: the meta-data on the project (e.g., the people involved and their roles), the software used with version number, the settings of the active learning model, the feature matrix used, records used for the first training dataset, the random seed values, and the intended stopping rule. It defines the starting point of the screening; thus, without it, the review will never be reproducible. These data will only have to be stored once, so they will not become huge. Also, these data can be pre-registered on generic platforms, such as the Open Science Framework, or specific platforms, such as Prospero.

### 5.3. Labeling Phase

5.3.1. Data Produced by the Annotator

In the labeling phase, the data produced by the annotator should be stored as completely as possible: the records that were labeled in the order they were labeled, the corresponding label, the screener (in case multiple screeners are labeling the same model), labeling time (so that the order of records can be retrieved), and any other actions by the annotator during the screening (e.g., changed decisions, notes added to a record). The data produced by the annotator will not get much bigger than the size of the input dataset and can be easily linked to the input data.

5.3.2. Data Produced by the Model

Most notably, the data produced by the model are where we need to strike a balance between reproducibility and storage size. Storing all the model data will result in enormous file sizes, whilst storing no model data means that we cannot reproduce the model's behavior.

For each iteration of the active learning cycle, we believe it is essential to store two components: the type of model that was used (instead of the model itself) and information to reconstruct the training set of labeled records on which the model was trained. With

this information, it is transparent what information has been used in each iteration of the active learning cycle and it is possible to recompute the relevance scores at any point in the active learning cycle. For deterministic models, or for probabilistic models where the random seed is stored, it is possible to recompute these scores exactly. The amount of time this takes is the same as the amount of time it took to train the original model and compute the scores.

Similarly, we do not need to store all the records used in the training set for each iteration. The model will be trained again on each iteration's current set of currently labeled records. Since we can already find which records were labeled in the annotator data and in which order, we do not need to store this information again during the model training. Instead of storing all the records in the training set, it is enough to store the number of records. For example, if we know that there are 50 records in the training set, then the training set consists of the first 50 labeled records. We can find exactly which 50 records these are in the annotator data. If the model data are stored in this way, the size will depend linearly on the size of the input dataset. This is much better than the quadratic scaling one would obtain by naively storing all the model data. Moreover, using the same hardware, reconstructing the model of a specific iteration takes approximately the same time as it took to train the original model. If we also stored the random seeds used in any of the probabilistic training algorithms, then we can almost exactly reproduce the original model. Most certainly, we will be able to reproduce the ranking produced by the original model. It turns out that this amounts to surprisingly little extra information since we already stored the data from the set-up phase and all data produced by the annotator.

Users may opt to store all the models learned during the AL process. While these models can often be reconstructed from other stored data, direct storage of the models can facilitate ease of reproducibility, especially in cases where computational resources are limited or when strict reproducibility is required for scrutiny or detailed analysis. We suggest software creators offer this as an optional feature, allowing users to make an informed choice based on their specific needs and constraints.

*5.4. Output Data*

The final output data should consist of two parts. Firstly, there is the list of labeled records with the corresponding label: seen and relevant, seen and irrelevant, and unseen. This is the file that will be used in the post-processing stages of a systematic review. Secondly, there is a technical file containing everything necessary for the reproducibility of the process. It contains everything we described above.

## 6. Reproducibility and Data Storage for Active Learning-Aided Systematic Screening—The Checklist

*6.1. RDAL Checklist*

The recommendations result in the RDAL Checklist (Reproducibility and Data storage for Active Learning-Aided Systematic Screening Checklist), which helps users and creators of active learning software make their screening process reproducible. Table 2 provides an easy-to-use checklist for screeners, collaborators, reviewers, or editors. The RDAL checklist can be used as an add-on to the PRISMA checklist specifically for systematic reviews implementing active learning in the screening phase. We advise publishing items 1–7 before data collection in a pre-registration, and storing items 1–14 on a general-purpose and domain-specific data repository under an open data license, like CC-BY 4.0. For an application of the checklist to the ASReview software, see Appendix A.

**Table 2.** The RDAL Checklist.

| Item | Type of Data | Available? | Stored Where? |
|---|---|---|---|
| | Pre-registration | | |
| 1. | Meta-data of the project (title, description, contact person, etc.) | | |
| 2. | Proces: (a) The team involved in screening with roles assigned, (b) how they will collaborate in the project, and (c) how they will use the AL-aided pipeline | | |
| 3. | Inclusion and exclusion criteria | | |
| 4. | Software plus version | | |
| 5. | Intended model(s): (a) feature extraction, (b) classifier, (c) query strategy, (d) balance strategy | | |
| 6. | The intended stopping rule | | |
| 7. | Training data: (a) The selection process of the records used for prior knowledge, (b) their labels | | |
| | Input data | | |
| 8. | Data: (a)_ Persistent object identifiers and (b) the texts used for screening | | |
| | Output data | | |
| 9. | All records, including the labels 'seen and relevant', 'seen and irrelevant', and 'unseen'. | | |
| | Technical data | | |
| 10. | Feature Matrix (or matrices if multiple feature extraction techniques are used) | | |
| 11. | Random seed values | | |
| 12. | Order of labeling and any changes in labeling decissions | | |
| 13. | Actual used model per iteration (if different across iterations. Otherwise, storing it once is enough): (a) feature extraction, (b) classifier, (c) query strategy, (d) balance strategy | | |
| 14. | Information about which record(s) were used to train which model | | |
| 15. | Model output (This item is optional [1]) | | |

[1] While we suggest storing other items on the list, this one is not recommended for storage because it consumes excessive space relative to its utility. As the primary purpose of the screening is not to gather model-related information, this item is not critical for reproducibility in the case of applying active learning for systematic screening of literature. You may choose to store it if space is not a concern or if you plan to use the information later. However, the other items are far more important for reproducibility.

Non-reproducibility in systematic reviews, particularly those employing active learning, often stems from the lack of detailed recording of decisions made during the annotation and model training processes. The RDAL Checklist addresses this challenge by providing a structured approach to capture essential information that might otherwise be overlooked or inconsistently recorded. By ensuring that all pertinent details, such as model settings, annotation criteria, and decision points, are consistently documented, the checklist plays a pivotal role in enabling other researchers to replicate the study with fidelity. This systematic approach to documentation not only enhances the reproducibility of the research but also contributes to its transparency and reliability.

Moreover, this checklist provides the developers of active learning software with a reference for what their software should be storing and including in the result. If the software has the option to easily export all the necessary data for a reproducible screening phase, it will facilitate users to report on their active learning process. This, in turn, gives more confidence in the software as a whole.

### 6.2. Example of the Technical Data

To provide an example of how the data produced by the human and by the model can be stored (item 9 in combination with items 12–14 of the checklist), we provide an illustration based on the output of the open-source software ASReview v1.0. For each iteration, ASReview stores the number of records in the training set and the type of model used.

We assume one screener is active, only one model is trained at a time starting with the default settings of ASReview (TF-IDF + Naïve Bayes), and the user wants to switch to a different model (sBert + Neural Network) after screening ten records. The screener starts with two records as prior knowledge; see the value '−1' for the training set size in the first two rows in Table 3. Note that any size of prior knowledge can be used. If you used a model that does not use prior knowledge, this would be indicated by a '0' in the table. After importing the data and selecting the prior knowledge, you have to wait for the text to be transformed into a format the model can understand using a feature extraction method. Using the feature matrix, a first classifier is trained (denoted by M1), which produces relevance scores. The most likely relevant record is shown to the annotator (first in the queue of M1), who assigns the label '1'. This label is based on a model with a training set size of 2.

**Table 3.** Slice of the model data.

| | Labeling Data | | | | Training Data | | |
|---|---|---|---|---|---|---|---|
| | Row Number | Record Identifier | Label | Labeling Time | Training Set Size | Record from Queue | Model |
| | 1 | 145 | 0 | 2022-12-23 09:32:08.46 | −1 | - | - |
| | 2 | 56 | 1 | 2022-12-23 09:32:40.13 | −1 | - | - |
| | Composing feature matrix (using TF-IDF) + training of 1st iteration of 1st model (NB) with two priors | | | | | | |
| *Screening Model I* | 3 | 120 | 1 | 2022-12-23 09:13:26.48 | 2 | 1st of M1 | TF-IDF + NB |
| | 4 | 442 | 0 | 2022-12-23 09:14:26.55 | 2 | 2nd of M1 | TF-IDF + NB |
| | 5 | 247 | 1 | 2022-12-23 09:15:27.11 | 3 | 1st of M2 | TF-IDF + NB |
| | 6 | 491 | 1 | 2022-12-23 09:16:28.43 | 4 | 1st of M3 | TF-IDF + NB |
| | 7 | 102 | 0 | 2022-12-23 09:16:28.59 | 5 | 1st of M4 | TF-IDF + NB |
| | 8 | 243 | 0 | 2022-12-23 09:17:30.12 | 5 | 2nd of M4 | TF-IDF + NB |
| | 9 | 401 | 0 | 2022-12-23 09:18:40.22 | 6 | 1st of M5 | TF-IDF + NB |
| | 10 | 279 | 0 | 2022-12-23 09:19:59.36 | 8 | 1st of M6 | TF-IDF + NB |
| | Model Switch | | | | | | |
| | Training of 1st iteration 1st model with sBert + NN (M7) with ten priors | | | | | | |
| *Screening Model II* | 11 | 4273 | 1 | 2022-12-27 09:00:47.83 | 10 | 1st of M7 | sBert + NN |
| | 12 | 366 | 0 | 2022-12-27 09:01:13.26 | 10 | 2nd of M7 | sBert + NN |
| | … | | … | … | … | … | |

In the implementation of ASReview, at most one model is being trained at any time. After the annotator makes a labeling decision, the program checks if there is no model training in which case a new model is triggered. In our example, this is done after the label is provided by the screener in row number 3 and the training of the model M2 is started using the labels of the first 3 rows. Instead of having to wait for M2 to finish training, the annotator already sees the fourth record, which was ranked second highest in M1. Assuming the annotator needs some time to read the fourth record, M3 has finished training. Thus, the fifth record presented is based on M2, etcetera.

In ASReview, the highest-ranked record is always selected to be shown to the annotator (if the query strategy is set to certainty-based sampling), even if the model is not finished with training. For the first set of records in our example, the training of the model was completed while the annotator was reading the abstracts. Therefore, the rank order could be updated before the human was finished reading, and they could always read the highest-ranked record. Now, assume the label of row number 7 is provided in a split-second (maybe because the title directly indicates the record is irrelevant) and the new model is not yet finished training. In this case, the next record shown to the annotator is the second in the queue of the previous model (i.e., M4) instead of the highest-ranked record of the new model (i.e., M5). After the labeling decision of row 9 has been made, the 10th row is estimated with 8 records as training data.

As shown by Teijema et al. [37], many light classifiers only take seconds to produce new results, but neural networks might take (much) longer. For slower models, it might take longer before the model can update the rank order, and the screener will see the second highest-ranked record of an older model until a new model is finished training and the rank order is updated again. Therefore, after labeling ten records, the annotator decides to switch to a different model based on a neural network architecture; see rows 10 and further.

Storing the information in the columns 'record identifier', 'label', 'labeling time', and 'training set size', plus the information about which model was used in which iteration, is enough to retrain the model at any point in time without having to store too much data.

## 7. Discussion

With this paper we aim to provide foundational insights into reproducibility and data storage within the context of screening prioritization using AI during the screening phase. However, there are also some clear boundaries to what we investigated. We recognize that other crucial phases of systematic reviews, such as database searching and data extraction, as well as the performance and biases of AI-aided screening, are beyond the scope of this study. These limitations highlight important avenues for future research.

One significant area that merits deeper investigation is the potential for biases introduced by AI tools used in the screening process. While we have outlined procedures to enhance transparency and mitigate some risk of bias, the complete evaluation of bias, particularly how AI influences the selection and exclusion of studies, remains to be addressed. Future work should focus on developing methodologies to effectively detect and correct for these biases. For instance, the NLF procedure (Noisy Label Framework) can be applied to address noisy labels, the inaccuracies in labels provided by experts, enhancing the overall reliability of the screening process [47]. Similarly, the SAFE procedure, a heuristic for deciding when to stop screening, includes a quality check to correct for potential false exclusions [48]. Nonetheless, biases will be inherent in any screening process, be it fully human or AI-aided. This is one more reason it is important to store data related to a review in an accessible and reusable manner, so that biases of finished reviews can be investigated.

Moreover, in future studies, it will be crucial to explore more robust methods for assessing inter-rater reliability (IRR) in the context of AI-aided systematic reviews [49]. Traditional metrics like Cohen's Kappa are often challenged by the data's nature in these reviews, specifically when data are missing not at random (MNAR) due to selective paper exclusion by AI mechanisms. Alternative metrics are proposed and designed to accommodate the complexities introduced by MNAR conditions, e.g., [50], which may provide more accurate assessments of IRR in AI-enhanced research settings.

In our study, we referred to the use of inclusion and exclusion criteria; however, it is important to note the distinct roles these criteria might play at various stages of the systematic review process. Inclusion criteria are typically applied during the preliminary screening phase to select studies based on predefined attributes. Conversely, exclusion criteria might be used during the same phase to exclude studies based on attributes such as publication year, language, and article type. However, exclusion criteria can also be applied to further refine and finalize the selection of relevant studies after the screening phase.

## 8. Conclusions

Text reading, processing, and screening require enormous amounts of time in almost any professional organization. With the emergence of online publishing, the number of scientific papers on any topic is skyrocketing. The COVID-19 crisis has illustrated how crucial and critical it can be to develop fast but rigorous systematic overviews of the literature. At the same time, accurate information retrieval is essential to provide access to facts. Orchestrated campaigns are spreading untruths, disinformation, mal-information, and misinformation, which are often unwittingly shared on social media. This raises questions about the quality, impact, and credibility of researchers, the judiciary, physicians, journalists, media networks, governments, and all other professionals, institutions, and networks acting as agents of knowledge. Therefore, reproducible and verifiable methods are needed to identify, select, and critically appraise all relevant decisions in the process of selecting textual data. Transparency will help increase quality and accountability to gain public trust.

Addressing the issue of reproducibility is a complex task, especially in the context of AI-aided systematic screening. The primary aim of our checklist is to address the gaps present in the PRISMA checklist when using AI-aided screening tools. It specifies the essential information that should be provided to ensure reproducible results. The checklist guarantees that all data inputs and outputs within the AI-aided screening process are systematically recorded and made accessible. This allows the same data to be used by others to replicate the study outcomes. We provide clear documentation of the algorithms, model parameters, and computational processes used, ensuring that the methods can be precisely replicated by other researchers. The checklist advocates for the transparency of both interim and final screening results, including the classification of data by AI tools. This ensures that results can be consistently replicated across studies using the same methodologies and data. By ensuring the reproducibility of data, methods, and results, our checklist indirectly supports the reproducibility of conclusions, provided that the analysis is consistently applied.

However, we need to strike a balance between data storage and reproducibility. On the one hand, storing the starting point is not enough to call the review reproducible. On the other hand, storing everything is overkill because not all of the enormous amount of information produced by the model is relevant. Where on this spectrum an AL-aided review pipeline should be, depends on the specific application.

There is a scale on which we can measure the software used for AL-aided screening and the needs of the users of this software. On the one side of this scale is the software that only provides the main output. The steps in between remain a black box because nothing gets reported on them. Users of this software should only be interested in the final results and not in reporting on the process of how these results were obtained. Software on the other side of the scale stores absolutely everything about the whole screening process. As we discussed, this can lead to considerable data storage sizes. For users on this side of the scale, it should be imperative to account for all the steps in the screening process, or the storage size is just not an essential factor for these users.

For classical reviews, the PRISMA guidelines go far toward storing everything. However, during pre-processing, the guidelines require saving the search string but not the actual screening input dataset. For AL-aided reviews, they shift away from storing everything since they only ask to report on which software was used and how it was used. They do not ask to store any information on the models that were used. As we noted, this means that we cannot reconstruct how the decisions of the model were made.

Users should think about where on the scale they want to be. How important is it to be able to report on the artificial intelligence models? Does this influence the trust in the conclusions? Does the storage size of the output matter? Similarly, software manufacturers should consider where on the scale they want to be. What is the intended group of users? How flexible is the software in reporting more or less of the detail of the process and saving more or less storage? We provided one storage model that goes further than

PRISMA does right now and can be flexibly extended, but everyone should assess their own considerations.

**Author Contributions:** Conceptualization, J.d.B., P.L. and R.v.d.S.; methodology, J.d.B., P.L. and R.v.d.S.; software, P.L. and J.d.B.; validation, P.L., J.d.B. and R.v.d.S.; formal analysis, P.L.; investigation, P.L., J.d.B. and R.v.d.S.; resources, J.d.B.; data curation, n/a; writing—original draft preparation, P.L.; writing—review and editing, R.v.d.S. and J.d.B.; visualization, P.L.; supervision, n/a; project administration, R.v.d.S.; funding acquisition, R.v.d.S. All authors have read and agreed to the published version of the manuscript.

**Funding:** The first author was funded by a grant from the European Commission, call H2020-INNOSUP-2020-02, under Grant Agreement ID 957029. The last author was funded by a grant from the Dutch Research Council under grant no. 406.22.GO.048.

**Institutional Review Board Statement:** Not applicable.

**Informed Consent Statement:** Not applicable.

**Data Availability Statement:** Not applicable.

**Acknowledgments:** This manuscript was compiled utilizing an array of large language models to optimize the writing process and ensure grammatical quality. For enhanced writing speed, we used Grammarly for language error correction, while OpenAI's GPT-4 was instrumental in refining sentences. The intellectual property (IP) contained in this document belongs exclusively to the authors. The AI-based tools mentioned were utilized solely as accelerators to enhance the writing process, assisting with speed and accuracy. They in no way contributed to the original ideas, insights, or intellectual content contained herein.

**Conflicts of Interest:** We would like to inform you that the authors of this manuscript are part of the ASReview project, which is an open-source and research-oriented software project for active learning-aided systematic literature reviews. However, we want to stress that there is no conflict of interest with the content of this paper. The ASReview software is freely available for use by researchers and we do not receive any financial or other benefits from the use of the software. This paper is a purely academic contribution that aims to improve the reproducibility of active learning-aided systematic reviews. We would also like to inform you that the first author of this manuscript, Peter Lombaers, is affiliated with IDfuse, a commercial company. IDfuse did not have any involvement in the funding or execution of the project. Furthermore, the content of this paper does not promote any commercial interests or products of IDfuse.

## Appendix A

**Table 1.** The RDAL checklist as applied by a person who performs a systematic review following the PRISMA guidelines. The person publishes a pre-registration and uses the open-source software ASReview v1.6.2 for the screening phase. After finishing the screening they export two files from the software: an excel file 'review.xlsx' containing the main output of the review, and a technical file 'review.asreview'.

| Item | Type of Data | Available as Output of the Software? | Stored Where? |
|---|---|---|---|
| | | Pre-registration | |
| 1. | Meta-data of the project (title, description, contact person, etc.) | Yes | In the file *project.json* inside the technical file *'review.asreview'*. |
| 2. | Proces: (a) The team involved in screening with roles assigned, (b) how they will collaborate in the project, and (c) how they will use the AL-aided pipeline | No | This information is not stored by the software but is part of the pre-registration. |
| 3. | Inclusion and exclusion criteria | No | This information is not stored by the software but is part of the pre-registration. |
| 4. | Software plus version | Yes | In the file *project.json* inside the technical file *'review.asreview'*. |
| 5. | Intended model(s): (a) feature extraction, (b) classifier, (c) query strategy, (d) balance strategy | Yes | In the file *settings_metadata.json* inside the technical file *'review.asreview'*. |
| 6. | The intended stopping rule | No | This information is not stored by the software but is part of the pre-registration. |
| 7. | Training data: (a) The selection process of the records used for prior knowledge, (b) their labels | Partly | The identifiers and labels of the records used for prior knowledge is available as part of the output file (a separate column for tabular data and a flag in the note field for RIS files). The information of the selection process is not stored by the software but is part of the pre-registration. |
| | | Input data | |
| 8. | Data: (a) Persistent object identifiers and (b) the texts used for screening | Yes | Available in the export file 'review.xlsx'. |

**Table 1.** *Cont.*

| Item | Type of Data | Available as Output of the Software? | Stored Where? |
|---|---|---|---|
| | | Output data | |
| 9. | All records, including the labels 'seen and relevant', 'seen and irrelevant', and 'unseen'. | Yes | Available in the export file 'review.xlsx'. |
| | | Technical data | |
| 10. | Feature Matrix (or matrices if multiple feature extraction techniques are used) | Yes | Stored as part of the technical 'review.asreview' file. |
| 11. | Random seed values | No | The seed values are not stored by the software. One can start the software with pre-determined seed values, but this value is not stored using this version of the software. |
| 12. | Order of labeling and any changes in labeling decisions | Yes | Stored as part of the technical 'review.asreview' file. |
| 13. | Actual used model per iteration (if different across iterations otherwise, storing it once is enough): (a) feature extraction, (b) classifier, (c) query strategy, (d) balance strategy | Yes | Stored as part of the technical 'review.asreview' file. |
| 14. | Information about which record(s) were used to train which model | Yes | Stored as part of the technical 'review.asreview' file. |
| 15. | Model output (This item is optional) | n/a | In the interest of saving storage space, the user is not interested in storing the output of the model. In practice, the ranking for the last iteration of the model is stored as part of the technical 'review.asreview' file. |

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
