# Peer review of "Reproducibility and Data Storage for Active Learning-Aided Systematic Reviews"

_applsci, doi:10.3390/app14093842_

Round 1

Reviewer 1 Report

Comments and Suggestions for Authors

First of all, I would like to thank the editor(s) for considering me as a reviewer. Secondly, I would like to applaud the authors for an interesting proposition or rather a conceptual checklist that aids reproducibility for active learning-based systematic reviews. I would say that the paper is indeed interesting and, in somewhat academic circles, necessary. However, right off the bat, I believe that the paper should provide more context and intention, or target groups that you are trying to reach. Nevertheless, my pointers, suggestions and recommendations for improvement are in the following.

1. Introduction

I would argue that the background and rationale for starting your manuscript should entail more emphasis on what type of "systematic reviews" are you tackling. For instance, when you're saying a "Systematic review" do you mean vertical (e.g., Scoping, Narrative, Mapping, or even Umbrella Systematic Reviews) or vertical (e.g., Systematic Literature Review and Meta&Analysis, Diagnostic Test Accuracy Meta-Analysis, Network Meta-Analysis)? (see: 10.1111/j.1471-1842.2009.00848.x) Also, in line number 44 you say systematic review (or meta-analysis), these are completely two different things. You can conduct a meta-analysis study without using systematic review and vice versa. The systematic review is mostly considered as a protocol that will assure transparency and replicability of a study. Also, your argument that scientific output doubles every nine years is obsolete. In information and library science, this goes much faster, not to mention the medical and health spheres. Consider rather this - "...the number of systematic literature reviews will double by 2025" (see: 10.1007/s11192-024-04935-2).

The Box 1. "The process starts with a dataset of unlabeled records with meta-data containing titles and abstracts..." Does not contain keywords? "Next, an active learning model needs to be selected, including a feature extraction technique, a classification algorithm (i.e., a machine learning model), a query strategy, and a balancing strategy to deal with the extremely imbalanced number of relevant records in the data." Relevant or irrelevant records?

2. Reproducibility in the context of systematic screening

Tackling the issue of reproducibility is an interesting task; however, I must ask, are you considering attacking the notion of reproducibility of screening or replicability of screening? "Reproducible research is not an easy concept to define". I would strongly agree with this, but, as stated, what do you want to obtain from this? Reproducible (replicable) data? Reproducible method? Reproducible screening results? Reproducible findings? More importantly, are you tackling the issue of bias in this case? If so, there are plenty of qualitative methods (e.g., GRADE-CERQual - https://www.cerqual.org/) for different types of systematic reviews that can be utilised for this. Maybe you can consider their frameworks in the quantitative data domain.  

A note: Amid the reading I came across a thought that can potentially aid and/or help you validate or in some instances add reliability to the metric of reproducibility. Did you consider adding some statistical tests to measure the association between the rater's scores (e.g., Cohens Kappa? or Fleiss Kappa?)? Also, I would argue that inclusion/exclusion criteria can be considered formulaic. Sometimes only inclusion criteria are used (pre)screening phase, for instance, imposing criteria of period, language, article type (primary original studies), full-text articles only (not editorials, communications), etc. In contrast, exclusion criteria can be imposed during or post-screening phase when deciding what to include in the systematic review. Thus, maybe just place "Isolation criteria" in the preprocessing. This way you impose what articles are you isolating from the corpus or scientific discourse.

Given the argumentation scheme, mostly following the ideology of Goodman Fanelli [34], I would say I would strongly disagree with this qualitative analogy, especially with the last statement (line numbers 260-262) - "inferential reproducibility". Namely, "...gives confidence that missing a specific record is not essential for the final conclusions...", could not be more in disagreement with the statement, especially since in my experience when conducting meta-analysis (or meta-synthesis) one study could change it all. Looking at the reference you've proposed [35] I agree with their fact that missing or excluding 5-10% of "last-to-find records" will not impact the outcome since in my prior research experience, indeed, even as extreme as the last 20% of screened records (commonly in using Herzing's publish or perish software) may not contain relevant study to be incorporated in the corpus of relevant studies, but your description "missing a specific record is not essential for the conclusion" seems a bit off in that description and it gives more impression that you emphasise that missing an article will not impact the conclusion. Maybe consider changing these sentences. 

3. Data in AL-aided screening

No remarks are to be added.

4. A trade-off between reproducibility and data storage

No remarks are to be added.

5. Data storage recommendations for systematic screening

No remarks are to be added.

HINT: I would like to applaud the authors for the well-articulated and presented analysis through the previous sections. Also, I do not have any positive (or negative) critique that will help you increase the quality of the manuscript since I was more in "awe of the articulation" you provided, and I do not feel competent enough to provide you with an assessment on the proposed sections. But I hope someone will help you improve (if necessary) the aforementioned sections.

7. Conclusions

Seeing that a clear and articulated description of your work and your contribution to the literature is presented, I only have two questions. What would you consider were the limitations of your work? Can data quality metrics be used in your checklist?

Again I would like to thank the editor(s) for considering me as a reviewer. As far as the suitability of the publication of the manuscript, I would recommend minor revisions. The paper is well-articulated, well-presented and interesting to the audience.

Author Response

First of all, I would like to thank the editor(s) for considering me as a reviewer. Secondly, I would like to applaud the authors for an interesting proposition or rather a conceptual checklist that aids reproducibility for active learning-based systematic reviews. I would say that the paper is indeed interesting and, in somewhat academic circles, necessary. However, right off the bat, I believe that the paper should provide more context and intention, or target groups that you are trying to reach. Nevertheless, my pointers, suggestions and recommendations for improvement are in the following.

Reply:

Thank you for your encouraging remarks and constructive feedback. We appreciate your recognition of the conceptual checklist's potential contribution to the field of active learning-based systematic reviews. We agree that providing additional context and specifying target groups could enhance the paper's clarity and applicability. We will address these aspects as per your suggestions in the revised manuscript.

---

1. Introduction

I would argue that the background and rationale for starting your manuscript should entail more emphasis on what type of "systematic reviews" are you tackling. For instance, when you're saying a "Systematic review" do you mean vertical (e.g., Scoping, Narrative, Mapping, or even Umbrella Systematic Reviews) or vertical (e.g., Systematic Literature Review and Meta&Analysis, Diagnostic Test Accuracy Meta-Analysis, Network Meta-Analysis)? (see: 10.1111/j.1471-1842.2009.00848.x) Also, in line number 44 you say systematic review (or meta-analysis), these are completely two different things. You can conduct a meta-analysis study without using systematic review and vice versa. The systematic review is mostly considered as a protocol that will assure transparency and replicability of a study.

Reply:

Thank you for your insightful observations concerning the importance of clearly distinguishing between different methodologies and their unique characteristics. In the revised text, we have aimed to clarify that we refer to the outcomes of systematic searches rather than the post-processing phase. We have explicitly mentioned that the results from systematic searches can be utilized for different forms of systematic reviews, including scoping, narrative, mapping, umbrella systematic reviews, as well as different types of meta-analyses such as diagnostic test accuracy and network meta-analysis; and we have added the reference suggested by the reviewer. We also carefully checked the manuscript to distinguish between systematic reviewing (which we sometimes used incorrectly) and systematic screening. We hope this revision better addresses the need for clarity and specificity that you pointed out. 

---

 Also, your argument that scientific output doubles every nine years is obsolete. In information and library science, this goes much faster, not to mention the medical and health spheres. Consider rather this - "...the number of systematic literature reviews will double by 2025" (see: 10.1007/s11192-024-04935-2).

Reply:

We appreciate your suggestion and have incorporated the newer statistic provided in the reference you recommended. In the revised manuscript, we have updated this section to reflect more current and specific data concerning the growth of systematic literature reviews, as suggested.

The Box 1. "The process starts with a dataset of unlabeled records with meta-data containing titles and abstracts..." Does not contain keywords? "Next, an active learning model needs to be selected, including a feature extraction technique, a classification algorithm (i.e., a machine learning model), a query strategy, and a balancing strategy to deal with the extremely imbalanced number of relevant records in the data." Relevant or irrelevant records?

We improved the text of Box 1 to clarify the text and to add the option of using keywords. 

---

2. Reproducibility in the context of systematic screening

Tackling the issue of reproducibility is an interesting task; however, I must ask, are you considering attacking the notion of reproducibility of screening or replicability of screening? "Reproducible research is not an easy concept to define". I would strongly agree with this, but, as stated, what do you want to obtain from this? Reproducible (replicable) data? Reproducible method? Reproducible screening results? Reproducible findings? 

Reply:

Thank you for highlighting the complexity of defining reproducibility, especially in the context of AI-aided systematic reviews. The primary aim of our checklist is to address the gaps present in the PRISMA checklist when using AI-aided screening tools. It specifies the essential information that software should provide to make the results reproducible.

Regarding the specific aspects of reproducibility: data, methods, results, and conclusions—yes, all should indeed be reproducible. The reproducibility of the results does not include the iterations of the model itself but pertains to the outcome they produce. Similarly, the conclusions are reproducible to the extent that the analysis leading to them is reproducible. This comprehensive approach ensures that each component of the screening process can be consistently replicated or reproduced, enhancing the reliability and validity of the research: 

  • Data Reproducibility: The checklist ensures that all data inputs and outputs within the AI-aided screening process are systematically recorded and made accessible, allowing the same data to be used by others to replicate the study outcomes.
  • Methodological Reproducibility: We provide clear documentation of the algorithms, model parameters, and computational processes used, ensuring that the methods can be exactly replicated by other researchers.
  • Results Reproducibility: The checklist advocates for the transparency of interim and final screening results, including how data was classified by the AI tools. This ensures that results can be consistently replicated across studies using the same methodologies and data.
  • Conclusions Reproducibility: By ensuring the reproducibility of data, methods, and results, our checklist indirectly supports the reproducibility of conclusions, provided the analysis is consistently applied.

We acknowledge the challenging task in the conclusion section, and we added the definitions stated above. 

---

More importantly, are you tackling the issue of bias in this case? If so, there are plenty of qualitative methods (e.g., GRADE-CERQual - https://www.cerqual.org/) for different types of systematic reviews that can be utilised for this. Maybe you can consider their frameworks in the quantitative data domain.  

Reply:

Thank you for raising the important issue of bias and we acknowledge that examining bias in the papers included in systematic reviews is crucial and typically forms part of subsequent phases of review processes. In the current scope of our manuscript, however, we focus primarily on establishing a reproducible and transparent process for data storage and active learning applications during the screening phase. We view the evaluation of bias in included papers as a subsequent step that falls outside the scope of our current paper.

However, the checklist we propose, "Reproducibility and Data Storage for Active Learning-Aided Systematic Reviews," is designed to facilitate transparency in the use of AI tools, which can assist in assessing potential biases of AI-aided methods used on review papers. Specifically, it aims to ensure that the interaction between machine learning models and human reviewers is clear, thereby mitigating the risk of introducing opaque, "black box" biases into the review process.

We added the following text to the introduction section to clarify the current focus of the current paper:

The current paper will start with a general discussion of reproducibility in the context of AI-aided systematic reviews. After that we fully focus on reproducibility for the phase of systematically screening records. We will not look further at other phases of a systematic review, such as database searching, or data extraction. AI-tools can be used in these phases, and reproducibility is important there as well, but it’s a topic for another paper. We will also not look at topics such as the performance of AI-aided screening, or biases in AI-aided screening. These in fact are motivations for making screening data reproducible and accessible.

Also, we added a new Discussion section to address the suggestion by the reviewer. 

---

A note: Amid the reading I came across a thought that can potentially aid and/or help you validate or in some instances add reliability to the metric of reproducibility. Did you consider adding some statistical tests to measure the association between the rater's scores (e.g., Cohens Kappa? or Fleiss Kappa?)?

Reply:

Thank you for your suggestion to incorporate statistical tests such as Cohen’s Kappa or Fleiss Kappa to measure the association between rater's scores and enhance the metric of reproducibility. Your point is well-taken; however, the use of these measures in the context of our study involves some complexities.

Our methodology already is valid for single-screener scenarios, which limits the applicability of inter-rater reliability (IRR). Of course, PRISMA requires multi-screener setups, where disagreements are deliberated, and labels are adjusted accordingly. As outlined by PRISMA standards, the IRR represents a crucial metric in studies involving subjective assessments to ensure that individual biases do not significantly influence the results. However, the use of active learning models introduces the complication of data being "missing not at random" (MNAR), which significantly impacts the reliability of traditional IRR measures like Kappa.

Specifically, in AI-aided systematic reviews, the study selection mechanism often excludes papers deemed irrelevant by the AI, creating an MNAR scenario where the data's missingness is directly related to its inherent characteristics. This can severely bias IRR estimates, leading to potentially misleading conclusions about the screening process's reliability.

Considering these issues, alternative metrics may be more suitable in AI-enhanced research environments, such as Gwet’s Kappa, the Regular Category Kappa, the sparse probability of agreement, and a chance-corrected agreement coefficient via a Bayesian framework, which better handles the complexities introduced by MNAR conditions.

However, this is outside the current scope of our paper. We did, however, include it in the new discussion section under future studies. 

---

 Also, I would argue that inclusion/exclusion criteria can be considered formulaic. Sometimes only inclusion criteria are used (pre)screening phase, for instance, imposing criteria of period, language, article type (primary original studies), full-text articles only (not editorials, communications), etc. In contrast, exclusion criteria can be imposed during or post-screening phase when deciding what to include in the systematic review. Thus, maybe just place "Isolation criteria" in the preprocessing. This way you impose what articles are you isolating from the corpus or scientific discourse.

Reply:

Thank you for your insightful comment regarding the use of inclusion and exclusion criteria in different phases of the screening process. We appreciate your suggestion and have included it in the limitations section of our paper to address potential improvements in methodology.

---

Given the argumentation scheme, mostly following the ideology of Goodman Fanelli [34], I would say I would strongly disagree with this qualitative analogy, especially with the last statement (line numbers 260-262) - "inferential reproducibility". Namely, "...gives confidence that missing a specific record is not essential for the final conclusions...", could not be more in disagreement with the statement, especially since in my experience when conducting meta-analysis (or meta-synthesis) one study could change it all. Looking at the reference you've proposed [35] I agree with their fact that missing or excluding 5-10% of "last-to-find records" will not impact the outcome since in my prior research experience, indeed, even as extreme as the last 20% of screened records (commonly in using Herzing's publish or perish software) may not contain relevant study to be incorporated in the corpus of relevant studies, but your description "missing a specific record is not essential for the conclusion" seems a bit off in that description and it gives more impression that you emphasise that missing an article will not impact the conclusion. Maybe consider changing these sentences. 

Reply:

Thank you for your critical feedback on our use of the term "inferential reproducibility" and the implications of missing specific records in the context of systematic screening. We recognize the importance of each study in potentially altering the conclusions of, for example, a meta-analysis and agree that our phrasing may have oversimplified this aspect. We have adjusted the text accordingly. 

---

3. Data in AL-aided screening

No remarks are to be added.

4. A trade-off between reproducibility and data storage

No remarks are to be added.

5. Data storage recommendations for systematic screening

No remarks are to be added.

HINT: I would like to applaud the authors for the well-articulated and presented analysis through the previous sections. Also, I do not have any positive (or negative) critique that will help you increase the quality of the manuscript since I was more in "awe of the articulation" you provided, and I do not feel competent enough to provide you with an assessment on the proposed sections. But I hope someone will help you improve (if necessary) the aforementioned sections.

Reply:

We are pleased to hear that the content resonated well with you. We appreciate your honesty in the assessment of our manuscript and value the encouragement to seek further improvements.  

---

  1. Conclusions

Seeing that a clear and articulated description of your work and your contribution to the literature is presented, I only have two questions. What would you consider were the limitations of your work? Can data quality metrics be used in your checklist?

Reply:

We have added a Discussion section in which we address these two questions (and more!).

---

Again I would like to thank the editor(s) for considering me as a reviewer. As far as the suitability of the publication of the manuscript, I would recommend minor revisions. The paper is well-articulated, well-presented and interesting to the audience.

Reply:

Thank you for your positive feedback!! 

---

Reviewer 2 Report

Comments and Suggestions for Authors

Dear Editor,

In this manuscript, the authors present a guideline, called RDAL-Checklist, for performing "Active Learning-Aided Systematic Reviews".

Systematic literature reviews (SLR) are important for summarizing the state of the art in a given field of research. They differ from traditional reviews in reducing the bias introduced by the individual performing the review. Furthermore, systematic reviews differ in that they are "reproducible", that is, if the initial definitions were followed, the final result can be reproducible by other individuals.

The literature contains several guidelines for performing SLRs (many of them are adapted to specific areas of activity). However, most are adapted from PRISMA (Preferred Reporting Items for Systematic Reviews and Meta-Analyses).

An important point in this work is that the authors address a variation of SLRs: Active Learning-Aided (AL) Systematic Reviews. AL reviews use artificial intelligence algorithms to support decision-making. This can solve one of the main problems of SLRs: the complexity of realization ("Artificial intelligence can help to speed up the process of searching through large amounts of text data"). 

Performing an SLR is costly since the ultimate objective is to summarize an entire area of knowledge. Furthermore, the process must be reproducible, so the review organizers may need to do a lot of work.

Artificial intelligence assistance can be useful but also bring new problems.

Below, I present some questions and concerns regarding this:

1—Firstly, this manuscript should not be classified as an original research article. The abstract does not make the objective of the article clear, nor did the experiments carried out. Therefore, I don't think it should be classified in that category.

We can compare how other related works have been published. For example, the Kitchenham guideline used in software engineering (see https://citeseerx.ist.psu.edu/document?repid=rep1&type=pdf&doi=29890a936639862f45cb9a987dd599dce9759bf5) and BiSLR used in bioinformatics (see http://dx. doi.org/10.13140/RG.2.2.28550.06721/1) were published as "Technical Reports".

Based on the MDPI/Applied Sciences journal standard, this manuscript should be submitted as a "Protocol" (see https://www.mdpi.com/about/article_types).

Therefore, I suggest submitting it in this category (see the necessary sections in the link above).

2 — I have a more generalized question about the proposal: How do you deal with the bias generated by the annotator? Can the algorithms used add new biases?

Systematic reviews aim to eliminate bias, but the "human component" will inevitably insert new biases into the text. However, this can be good from a certain point, as the review construction process is iterative, and the objectives may change slightly during its execution. When using artificial intelligence, the machine will define the best path, and this can strongly influence the final result. How does your method handle this?

3 — About the text:

Based on box 1:

Is Active Learning a semi-supervised learning strategy? In other words, does the researcher need to evaluate and label a set of articles so that the strategy can be adopted? Would it be this? I believe box 1 could explain the beginning of the process in more detail. Where do these labels come from?

EDIT: In line 303, section "3.2 Data in the labeling phase" this is finally explained. I then suggest reorganizing the presence of this box. Either remove it or improve the explanation.

Page 5—Box 2 was not announced before. It doesn't make sense that it was added there, so it is unnecessary.

In the excerpt (unnumbered line; box 2): "Classical active learning [REF] refers [...]" I believe a reference is missing in [REF], or is it an acronym?

Line 265 - the authors cite Table 1, but it only appears 2 pages later.

Table 1 - describe the acronym S/M/L

Author Response

 In this manuscript, the authors present a guideline, called RDAL-Checklist, for performing "Active Learning-Aided Systematic Reviews". Systematic literature reviews (SLR) are important for summarizing the state of the art in a given field of research. They differ from traditional reviews in reducing the bias introduced by the individual performing the review. Furthermore, systematic reviews differ in that they are "reproducible", that is, if the initial definitions were followed, the final result can be reproducible by other individuals. The literature contains several guidelines for performing SLRs (many of them are adapted to specific areas of activity). However, most are adapted from PRISMA (Preferred Reporting Items for Systematic Reviews and Meta-Analyses). An important point in this work is that the authors address a variation of SLRs: Active Learning-Aided (AL) Systematic Reviews. AL reviews use artificial intelligence algorithms to support decision-making. This can solve one of the main problems of SLRs: the complexity of realization ("Artificial intelligence can help to speed up the process of searching through large amounts of text data"). Performing an SLR is costly since the ultimate objective is to summarize an entire area of knowledge. Furthermore, the process must be reproducible, so the review organizers may need to do a lot of work. Artificial intelligence assistance can be useful but also bring new problems.

Reply:

Thank you for your comprehensive review and thoughtful analysis of our manuscript.

---

Below, I present some questions and concerns regarding this:

 1—Firstly, this manuscript should not be classified as an original research article. The abstract does not make the objective of the article clear, nor did the experiments carried out. Therefore, I don't think it should be classified in that category.We can compare how other related works have been published. For example, the Kitchenham guideline used in software engineering (see https://citeseerx.ist.psu.edu/document?repid=rep1&type=pdf&doi=29890a936639862f45cb9a987dd599dce9759bf5) and BiSLR used in bioinformatics (see http://dx. doi.org/10.13140/RG.2.2.28550.06721/1) were published as "Technical Reports".  Based on the MDPI/Applied Sciences journal standard, this manuscript should be submitted as a "Protocol" (see https://www.mdpi.com/about/article_types).  Therefore, I suggest submitting it in this category (see the necessary sections in the link above).

Reply:

Thank you for pointing out the issue with the classification of our manuscript. We agree that the content and structure may not meet the typical criteria for an original research article. Based on your feedback, we have reclassified the manuscript as a protocol paper.

---

2 — I have a more generalized question about the proposal: How do you deal with the bias generated by the annotator? Can the algorithms used add new biases?

Reply:

Thank you for your question on bias generated by the annotator. We have added a discussion section in which we also address potential biases from annotators. References to relevant studies and guidelines are included to support our approach.

 ---

Systematic reviews aim to eliminate bias, but the "human component" will inevitably insert new biases into the text. However, this can be good from a certain point, as the review construction process is iterative, and the objectives may change slightly during its execution. When using artificial intelligence, the machine will define the best path, and this can strongly influence the final result. How does your method handle this?

Reply:

Thank you for highlighting the inherent biases introduced by both human and AI elements in the systematic review process. As noted in our manuscript, while our current scope primarily focuses on setting frameworks for reproducibility and data storage, we acknowledge the importance of addressing biases. In our discussion section, we emphasize that biases are inevitable in any screening process, whether fully human or AI-aided.

---

3 — About the text:

Based on box 1:

Is Active Learning a semi-supervised learning strategy? In other words, does the researcher need to evaluate and label a set of articles so that the strategy can be adopted? Would it be this? I believe box 1 could explain the beginning of the process in more detail. Where do these labels come from?

Reply:

Thank you for your question. Yes, Active Learning involves initial labeling by researchers to train the model. We've updated Box 1 in our manuscript to clarify this process and the source of the labels.  

---

EDIT: In line 303, section "3.2 Data in the labeling phase" this is finally explained. I then suggest reorganizing the presence of this box. Either remove it or improve the explanation.

Reply:

We have updated the text according to the suggestion of the reviewer.

---

Page 5—Box 2 was not announced before. It doesn't make sense that it was added there, so it is unnecessary.

Reply:

We have added the reference to Box 2. 

---

In the excerpt (unnumbered line; box 2): "Classical active learning [REF] refers [...]" I believe a reference is missing in [REF], or is it an acronym?

Reply:

We have removed [REF] which was indeed a mistake.

---

Line 265 - the authors cite Table 1, but it only appears 2 pages later.

Reply:

We leave it to the typesetter to make sure to put Table 1 at the correct place in the final version and we will carefully check the page proofs. 

---

Table 1 - describe the acronym S/M/L

Reply:

We have added the definition of the acronym.

---

Reviewer 3 Report

Comments and Suggestions for Authors

I have read the paper and consider adequate for the journal.

The introduction could be better in terms of situating the contribution of the proposal.

The final discussion could be extended to discuss the current literature and the challenges ahead.

Author Response

I have read the paper and consider adequate for the journal.

Reply:

Thank you for your positive assessment

---

The introduction could be better in terms of situating the contribution of the proposal.

Reply:

We have revised and improved the introduction section to better highlight the contribution of our proposal.

---

The final discussion could be extended to discuss the current literature and the challenges ahead.

Reply:

We have added a new discussion section to comprehensively address the current literature and the future challenges.

---

Round 2

Reviewer 2 Report

Comments and Suggestions for Authors

Dear Editor, 

The authors have addressed most of my concerns. I believe the article can be published.

To the authors:

In several parts of the text (in particular, in lines 54-64, 88, 141), there are unspaced references to the text. Would it be possible to add spacing?

Extra suggestion: Could the supplementary materials include an example of Table 2 filled (with explanations)? This could serve as a guide for potential readers who want to implement your method.

Author Response

In several parts of the text (in particular, in lines 54-64, 88, 141), there are unspaced references to the text. Would it be possible to add spacing?

=> we have updated the spacing

---

Extra suggestion: Could the supplementary materials include an example of Table 2 filled (with explanations)? This could serve as a guide for potential readers who want to implement your method.

=> This has been an excellent suggestion because while filling in the table (see Appendix A) we also realized some items of the checklist could be improved.